


# Methane Sulphonic Acid in East Antarctic Coastal Firn and Ice Cores and Its Relationship with Chlorophyll-a and Sea Ice Extent

Emma Nilsson[1,2], Carmen P. Vega[3], Dmitry Divine[1], Anja Eichler[4,8], Tonu Martma[5], Robert Mulvaney[6], Elisabeth Schlosser[7], Margit Schwikowski[4,8,9], Elisabeth Isaksson[1]

[1]Norwegian Polar Institute, N-9296 Tromsø, Norway
[2]Department of Earth Sciences, Uppsala University, Uppsala, SE-75236, Sweden
[3]Subdepartamento de Climatología y Meteorología Aplicada, Dirección Meteorológica de Chile, Dirección General de Aeronáutica Civil, Portales 3450, Santiago, Chile
[4]Laboratory of Environmental Chemistry, Paul Scherrer Institute, 5232 Villigen PSI, Switzerland
[5]Institute of Geology, Tallinn University of Technology, Tallinn, Estonia
[6]British Antarctic Survey, Madingley Road, High Cross, Cambridge, Cambridgeshire CB3 0ET, UK
[7]Department of Atmospheric and Cryospheric Sciences (ACINN), University of Innsbruck, Innsbruck, Austria
[8]Oeschger Centre for Climate Change Research, University of Bern, 3012 Bern, Switzerland
[9]Department of Chemistry, Biochemistry and Pharmaceutical Sciences, University of Bern, 3012 Bern, Switzerland

*Correspondence to*: Emma Nilsson (nilssonemmax@gmail.com)

**Abstract.** Sea ice is important for both regional and global climate, but comprehensive sea ice records are lacking pre-1978, when global-scale spaceborne observations began. Attempts to reconstruct sea ice conditions in different regions of Antarctica with the help of methane sulphonic acid (MSA) records from ice cores have had varying success, highlighting the often-regional relationship between ice core MSA and sea ice. This study uses MSA records from three firn cores and one ice core drilled on Fimbul Ice Shelf in Dronning Maud Land, East Antarctica, to investigate the relationship to satellite-derived sea ice extent (SIE) in the Southern Ocean. Chlorophyll-a concentrations, serving as a measure of phytoplankton biomass, are correlated to the MSA records to further test the MSA – SIE relationship. The relationship to both SIE and chlorophyll-a differs largely between the different firn and ice core MSA records. We find significant correlations for the MSA records from the two higher accumulation core sites to SIE and chlorophyll-a in the Weddell Sea, Western Pacific Ocean, and Ross Sea Sectors. Furthermore, the use of stacked MSA records introduced significant correlations between MSA from the lower accumulation core sites and SIE. The absence of coherent correlation patterns between the MSA records across the four investigated cores and SIE or chlorophyll-a in the Southern Ocean suggests that the Fimbul Ice Shelf MSA records are not consistent proxies for regional SIE.

## 1 Introduction

Sea ice in the Southern Ocean exerts an important control on regional and global climate, atmospheric and oceanic circulation, as well as on local ecosystems (Nicol et al., 2000; Yuan and Martinson, 2000; Parkinson, 2004). Sea ice is also an important facilitator in producing dimethyl sulphide (DMS or $(CH_3)_2S$), one of the precursors to sulphate aerosols which





act as cloud condensation nuclei to cool the planet (Charlson et al., 1987; Watson and Liss, 1998; Vancoppenolle et al., 2013). Global-scale spaceborne observations of sea ice conditions only began in 1978. Further back in time, the sea ice

record can be reconstructed from whaling ship logs (Cotté and Guinet, 2007), but the spatial coverage is much smaller than what satellite imagery can provide. To reconstruct past sea ice beyond 1978, different sea ice proxies in paleoarchives are continuously being investigated and assessed. One potential sea ice proxy is methane sulphonic acid (MSA or $CH_3SO_3H$) preserved in Arctic and Antarctic ice cores. MSA is an oxidation product of DMS which in turn is produced in the oceans by phytoplankton, with the seasonal sea ice zone as an important source region (Curran and Jones, 2000). Wind transports MSA

from the marine boundary layer over open ocean to continental Antarctica, where it is deposited in the snow. DMS production is high during austral spring and summer in the seasonal sea ice zone (e.g., Kettle et al., 1999; Curran and Jones, 2000; Hezel et al., 2011), which forms the foundation of the pronounced seasonal variation of MSA concentration preserved in ice cores, allowing MSA to be investigated as a proxy for sea ice. In turn, that could allow more accurate reconstructions of past sea ice conditions prior to the historical observation period, which is vital for understanding paleoclimate and

projecting future climate change.

Chlorophyll-a concentrations in the near-surface ocean are reported by satellite ocean-colour data and are intimately connected to phytoplankton biomass, and therefore also DMS and MSA production. Subsequently, the connection between MSA and phytoplankton activity near the sea ice edge can be investigated using observed chlorophyll-a concentrations. In ice core studies, the relationship between chlorophyll-a concentration and MSA has been used to validate the MSA – sea ice

relationship (Thomas and Abram, 2016).

The relationship between MSA concentration in ice cores and sea ice extent (SIE) varies in previous studies. Studies in both East Antarctica (e.g., Curran et al., 2003; Foster et al., 2006) and West Antarctica (Abram et al., 2010; Thomas and Abram, 2016) have found significant positive correlations between MSA and SIE at a regional scale. Other research has found both positive and negative correlations, highlighting the importance of careful investigation of local wind transport

patterns which can modulate the MSA record in ice cores (Abram et al., 2007). Finally, some studies find no strong correlation between MSA and SIE at all, neither for MSA deposited in Antarctica (Hezel et al., 2011) nor for MSA concentration measured in the atmosphere (Preunkert et al., 2007). These varying results show that the ice core MSA – sea ice relationship is highly site dependent and that the suitability of each ice core MSA record as a sea ice proxy must be assessed separately. Furthermore, as there are now longer records of SIE and chlorophyll-a it is important to revisit the MSA

– SIE relationship.

In this study, the MSA records from three firn cores and one ice core from Fimbul Ice Shelf, Dronning Maud Land (Kaczmarska et al., 2004; Vega et al., 2016, 2018) are used to investigate the relationships between MSA, SIE, and chlorophyll-a in this region of Antarctica. The aim is to assess the potential usefulness of these MSA records as a sea ice proxy. To achieve this, the seasonality and resolution of the four FIS MSA firn and ice core records will be evaluated. Using

satellite records of SIE and chlorophyll-a concentrations the Southern Ocean, MSA – SIE and MSA – chlorophyll-a





correlations will be calculated. Finally, specific wind conditions prevalent over relevant oceanic areas as well as on FIS will be examined to assess the MSA variability in these records.

## 2 Methods

### 2.1 Study area

Fimbul Ice shelf (FIS) (Figure 1) is an ice shelf at the coast of Dronning Maud Land (DML), East Antarctica with an extent of 36 500 km$^2$. Several dome-shaped ice rises are situated on FIS. The firn cores used in this study have been collected from three ice rises called Kupol Ciolkovskogo (KC), Kupol Moskovskij (KM), and Blåskimen Island (BI). They are situated at a distance of approximately 200 km from each other, and their elevation ranges from 264 m a.s.l. (KC) and 268 m a.s.l. (KM) to 394 m a.s.l. (BI) (Vega et al., 2016). The S100 core site lies at an elevation of 48 m a.s.l. (Kaczmarska et al., 2004). KM,

BI, and S100 are located closer to the coast at 12 km, 10 km, and 3 km distance respectively, with the ocean as their northern boundary, whereas KC is situated 42 km from the coast (Table 1, Figure 1).

    Surface winds and precipitation at FIS are strongly influenced by low-pressure systems formed in the circumpolar trough. These cyclones north of the coast move east, causing easterly or north-easterly winds, which transport moist air onshore. Thus, coastal precipitation is mainly a product of these low-pressure systems (Schlosser et al., 2008). The precipitation

amount induced by the frontal systems during an event is determined by the temperature and humidity of the air masses, with air masses originating in polar latitudes resulting in less precipitation than lower latitude air masses. Generally, the Weddell Sea Sector is the main source region for moisture to DML (Noone et al., 1999; Reijmer and Broeke, 2001; Schlosser et al., 2008). During fair-weather periods, katabatic winds from southerly directions prevail.

    Another important factor influencing local wind and precipitation patterns is the topography of the ice shelf. Ice rises

constitute obstacles for the atmospheric flow, resulting in higher wind speeds and differences in precipitation at the ice rises (Lenaerts et al., 2014). The factors determining if there will be more precipitation on the lee side or windward side of the ice rise are the wind speed, the height of the ice rise relative to its horizontal extent, and the static stability of the air (Rotunno and Houze, 2007; Houze Jr., 2012). Furthermore, post-depositional redistribution influences the snow accumulation. The result of these significant spatial variations can be seen from the ice rises in this study having varying surface mass-balances,

with higher surface-mass balance at BI and KM than at KC (and S100) (Table 1) (Vega et al., 2016).



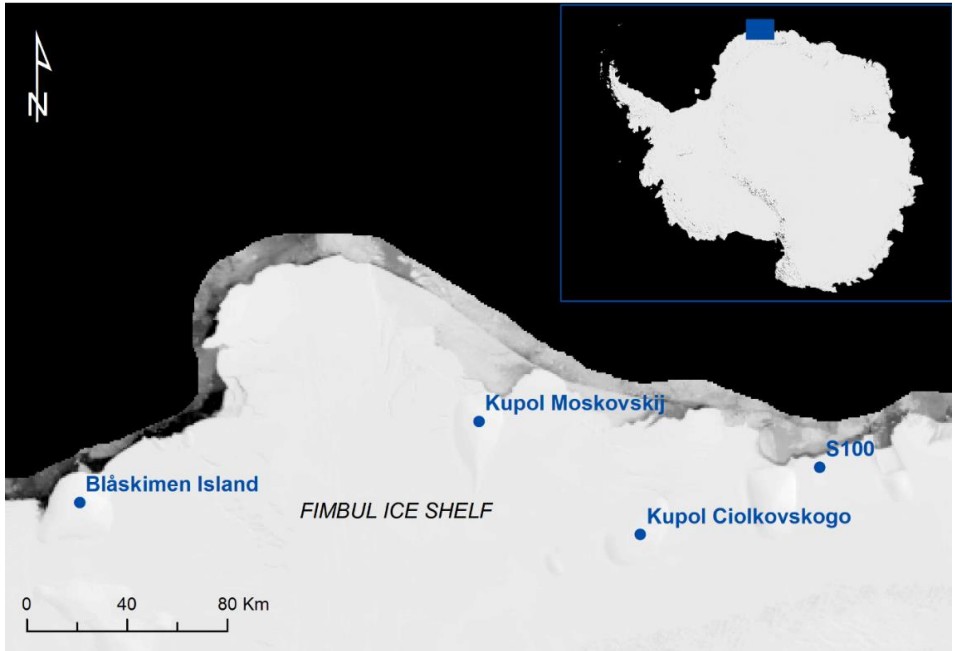

**Figure 1.** Map of Fimbul Ice Shelf and the location of the core sites Blåskimen Island (BI), Kupol Moskovskij (KM), Kupol Ciolkovskogo (KC), and S100. Inset map shows the location of Fimbul Ice Shelf on Antarctica. Satellite image is from MODIS Mosaic of Antarctica 2013-2014 (Haran et al. 2022).

## 2.2 Previous work: sampling and dating

During the field season 2000/2001, an ice core named S100 with a length of 100 m was retrieved from the eastern part of FIS as a part of the joint European Project for Ice Coring in Antarctica (EPICA) and the Norwegian Antarctic Research Expedition (NARE). In January 2012, a 20 m deep firn core was drilled on the KC ice rise on FIS, followed by two more 20 m deep firn cores drilled in January 2014 on the KM and BI ice rises respectively, during field expeditions of the Norwegian Polar Institute. The drill locations are identified in Figure 1.

The firn cores were sampled at a resolution of 4 to 8 cm, depending on the sample density and sample depth. Analysis of MSA and other major ions was conducted at Paul Scherrer Institute (PSI) (Vega et al., 2016, 2018). The S100 ice core was sampled at a resolution of 5 cm in the upper 6 m and 25 cm in the lower parts (Kaczmarska et al., 2004). Ice lenses and ice layers were identified and in the case of the KC core, the thickness of the lenses was measured (Vega et al., 2016).

Dating of the S100 core was performed by electrical conductivity measurements (ECM) and dielectric profiling (DEP) to identify volcanic horizons. High-resolution $\delta^{18}$O data sampled every 3 cm (Figure 2d) in the core were used to identify seasonal layers to further corroborate the volcanic peak counting method. The dating error for the S100 core was estimated as ± 3 years and represents the maximum difference calculated between dates of known volcanic eruptions and dates obtained through the counting of $\delta^{18}$O layers (Kaczmarska et al., 2004). The KC core was also dated using a combination of volcanic horizons and $\delta^{18}$O data but volcanic peaks in this case were identified from the non-sea-salt $SO_4^{2-}$ concentration.



Due to relatively low accumulation at the KC core site, the dating error was estimated to be ± 3 years (Vega et al., 2016). The KM and BI cores were dated through annual layer counting using seasonal cycles of $\delta^{18}$O data and major ion concentrations (Vega et al., 2016) (Figure 2). Under the assumption of constant, but different, precipitation rates during summer and winter an equidistant timescale was applied between summer maxima and winter minima. Because the accumulation rates are higher in the KM and BI cores compared to in the KC core, the seasonal variation in $\delta^{18}$O is better preserved, allowing a more accurate dating with an estimated error of ±1 year (Vega et al., 2016). The BI core spans the years 1996–2012, the KM core 1995–2012, the KC core 1958–2007, and the S100 core 1737–2000 (Table 1).

Further details about the retrieval and dating of the S100 core is described in Kaczmarska et al. (2004), whereas retrieval and dating of the three ice rises cores as well as major ion analysis of all four cores is described in Vega et al. (2016; 2018).

**Table 1.** Summary of the cores and coring locations from coastal DML used in this study **(Kaczmarska et al., 2004; Vega et al., 2016)**.

| Site | Elevation (m a.s.l.) | Surface mass balance (m w.e. yr⁻¹) | Distance from coast (km) | Core length (m) | Time coverage (years) |
|------|------|------|------|------|------|
| **BI** | 394 | ± 0.70 | 10 | 19.4 | 1996–2012 ± 1 |
| **KC** | 264 | ± 0.24 | 42 | 20 | 1958–2007 ± 3 |
| **KM** | 268 | ± 0.68 | 12 | 19.6 | 1995–2012 ± 1 |
| **S100** | 48 | ± 0.30 | 3 | 100 | 1737–2000 ± 3 |

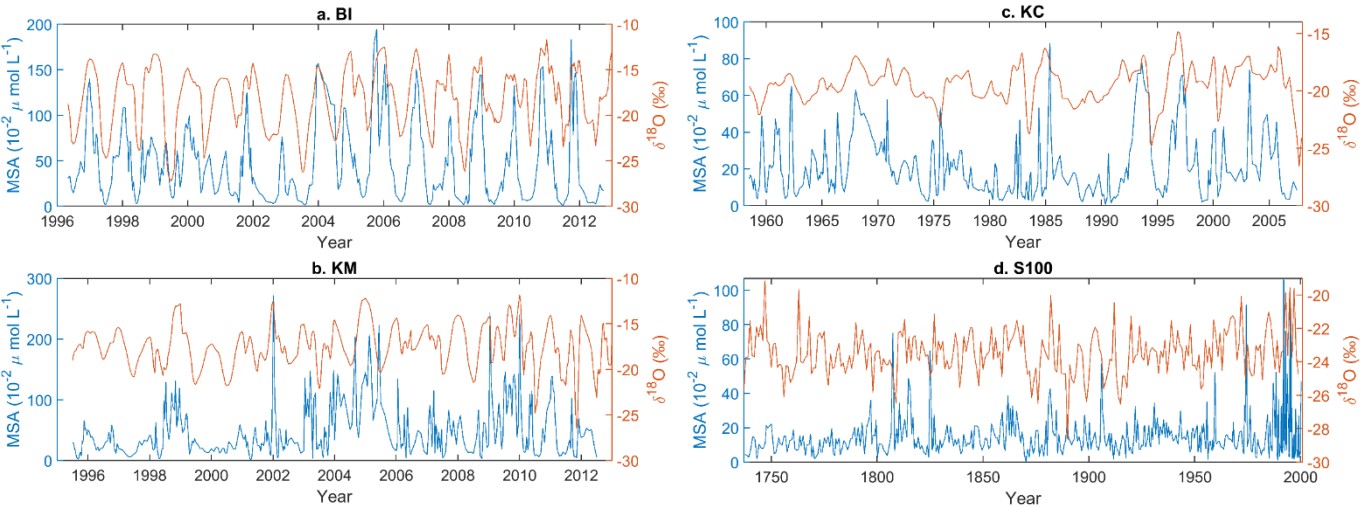

**Figure 2.** Seasonality of methane sulphonic acid (MSA) and δ18O in the three firn cores a. BI, b. KM, and c. KC from Vega et al. (2016), and in the ice core d. S100 from Kaczmarska et al. (2004). The MSA records are raw data.



### 2.3 Resolution of the MSA records

Ice rises are strongly influencing local meteorological patterns which affect snow accumulation (Lenaerts et al., 2014; Vega et al., 2016) and sea salt load (Vega et al., 2018). Since the majority of MSA is deposited through precipitation, the ice rises

will affect delivery of MSA to the core sites. The high snow accumulation at the BI and KM core sites compared to the S100 and KC core sites arises because of the symmetrical dome shape of the ice rises with slopes facing the incoming winds, forcing orographic precipitation as air flows past them (Vega et al. 2016). At the KC ice rise, the elongated ridge stretching from southwest to northeast causes north to north-easterly air flow to lift over a gentler slope, reducing the precipitation amount compared to BI and KM. As the S100 core is retrieved not from an ice rise but from the general ice shelf, the

accumulation patterns there are more like those of KC. The differences in accumulation at the core sites (Table 1) influence the temporal resolution in the cores. At the chosen sampling size, higher accumulation at the BI and KM sites allows sub-annual (seasonal) resolution, whereas the lower accumulation and subsequent lower resolution in the KC and S100 cores renders their MSA record not suitable for seasonal correlations to SIE and chlorophyll-a.

### 2.3 Statistical analysis

**2.3.1 MSA concentrations**

The records of MSA in the four previously dated firn and ice cores drilled on FIS used in this study were obtained via PANGAEA Data Publisher (Vega et al. 2022). The raw MSA data were converted into monthly values based on the provided decimal date. When working with seasonal MSA in the analysis, summer was set to encompass all months from October to March (ONDJFM) and winter spanned April to September (AMJJAS). The cut-off point between winter and

summer was set to the turn of the month between September–October because maximum SIE in the Southern Ocean usually occurs in mid- to late-September (EUMETSAT 2020).

A Wilcoxon rank-sum test was performed on populations of mean austral summer and winter concentrations in each core to assess if there is a statistically significant difference between summer and winter MSA concentrations. The null hypothesis of the test was that the medians of all the summer and winter mean MSA concentrations are the same.

The monthly MSA data were found to be log-normally distributed. Therefore, a log-transformation was applied to the data. Annual mean values of MSA were calculated by computing the mean of all log-transformed monthly MSA values for twelve months starting in July and ending in June, centring the annual averages around summer when DMS emissions are highest (Kettle et al., 1999; Curran and Jones, 2000; Hezel et al., 2011). Summer mean values of log-MSA were also calculated for the BI and KM cores by averaging the monthly log-MSA data over the six summer months.

The MSA data were also tested for significant trends by applying a linear regression model to the annual averaged MSA values in each core over the years overlapping the SIE data. To examine possible temporal autocorrelation in the MSA data, MATLAB's *autocorr*-function was applied to the time series of annual and seasonal (BI and KM) MSA anomalies.



Finally, the log-MSA values in all cores were normalised by calculating a standardised z-score to allow stacking of the MSA data from multiple cores. The z-scores of log-MSA data in the BI and KM cores were added to create one stack of

summer values and one stack of annual values and the z-scores of log-MSA data in the KC and S100 cores were added to create an annual stack. As all four cores do not overlap temporally no stack of all MSA data could be created.

The log transformed MSA data were only used in the correlation analysis whereas the statistical tests were performed on the raw MSA data.

### 2.3.2 Sea ice extent

Records of monthly sea ice extent (SIE) for the Southern hemisphere from November 1978 to present were downloaded from the Sea Ice Index, National Snow and Ice Data Centre, which tracks changes in Arctic and Antarctic sea ice (Fetterer et al., 2022). The product is derived from Sea Ice Concentrations from Nimbus-7 SMMR and DMSP SSM/I-SSMIS Passive Microwave Data. Sea ice in the monthly extent data is defined as grid cells with an average ice concentration higher than 15 %.

Due to the regional nature of the connection between MSA and sea ice data, the analysis was performed based on individual sectors of the Southern Ocean (Figure 3), based on the work by Cavalieri and Parkinson (2008) and Parkinson and Cavalieri (2012). The available SIE data were already divided into the five sectors used in this study.

Winter mean SIE (AMJJAS) was calculated for each sea sector and the Southern Ocean as a whole. To test for significant trends in the SIE, a linear regression model was applied to the winter averaged SIE in each sea sector, over the years

overlapping with the MSA records (1979–2012). In each sea sector where a significant trend in the SIE exists the data were detrended by subtracting the linear trend from the observations. To test for autocorrelation in the data, MATLAB's *autocorr*-function was applied to the time series of annual and seasonal SIE anomalies.



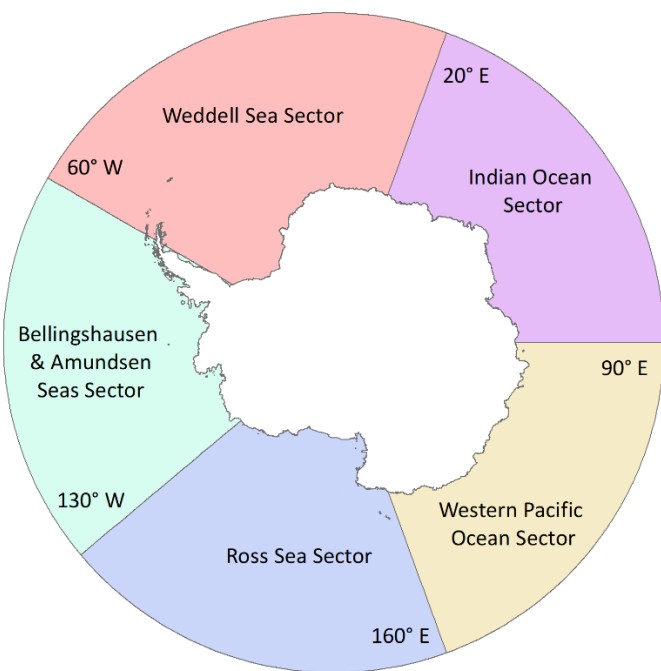

**Figure 3.** The five sea sectors used in this study and their longitudinal limits (Cavalieri and Parkinson 2008; Parkinson and Cavalieri 2012). The outline of Antarctica is provided by Gerrish et al., (2022).

### 2.3.3 Chlorophyll-a concentrations

The chlorophyll-a concentration data were derived from the MODIS instrument aboard the Aqua (EOS PM) satellite (NASA Goddard Space Flight Center, 2022). The data are mapped with a spatial resolution of 4 km. Annual and monthly chlorophyll-a concentrations for latitudes 50–90°S from 2002-07-04 to 2012-12-31 were downloaded.

To limit the chlorophyll-a data to only encompass the areas of the Southern Ocean influenced by sea ice, a sea ice mask was created. September maximum SIE was downloaded as shapefiles from the Sea Ice Index, National Snow and Ice Data Centre (Fetterer et al., 2022). Data for the years 2002–2012 were used to match the temporal coverage of the chlorophyll-a data. The September SIE shapefiles were processed in ArcGIS where the data for all the years were combined into one layer with the outer edge corresponding to the northernmost sea ice edge at each longitude occurring anytime over the years 2002-2012. Two buffer zones of 25 km and 100 km were then added around the maximum September sea ice edge-polygon. The sea ice mask was divided into five sectors with the same longitudinal limits as the SIE shown in Figure 3 (Cavalieri and Parkinson, 2008; Parkinson and Cavalieri, 2012), to enable sector-wise correlation also between chlorophyll-a and MSA. The chlorophyll-a data were then extracted based on the sea ice mask, with both the 25 km and 100 km buffer zone. From there, seasonal mean chlorophyll-a concentrations were calculated for the summer season, for the entire Southern Ocean and for each sea sector individually, with summer defined as ONDJFM. The seasonal chlorophyll-a concentrations were also tested for autocorrelation using MATLAB's *autocorr*-function.



### 2.3.4 Correlation analysis

The correlation analysis between log-MSA and, where applicable, detrended SIE was performed on one sea sector and core
at a time for the periods of overlap between the series. The seasonal and annual stacked log transformed MSA records were
also correlated to SIE. The MSA – SIE relationships were evaluated in MATLAB by fitting a robust linear regression model
to the data. When using a robust regression method each data point is weighted through iteratively reweighted least squares,
which reduces the sensitivity in the instance of non-Gaussian or heteroscedastic model residuals commonly found in ion
concentration data (Vega et al., 2018). After fitting a linear regression model, the Cook's distance on each observation was
evaluated to investigate potential outliers in the data. Any observation with Cook's distance larger than five times the mean
Cook's distance was removed from the data before re-fitting a robust regression model. This means that any observation
with a very high influence on the fitted response values was removed. The fit was then evaluated by looking at the model
output adjusted $R^2$-value and p-value. When calculating the seasonal correlations, winter sea ice was correlated to log-MSA
from the following summer. Correlations between winter sea ice and annual MSA centred on the following summer (i.e., a
year running from July to June) were also calculated for each core and sea sector combination.

For the correlation between MSA and chlorophyll-a, the calculated summer chlorophyll-a mean values for each sea
sector as well as the entire Southern Ocean were compared to summer and annual summer centred log-MSA in the BI and
KM cores, as well as the stacked summer and annual BI and KM log-MSA records using a robust linear regression. Any
observation with Cook's distance larger than five times the mean Cook's distance was removed from the data before re-
fitting a robust regression model. Since the temporal coverage of the S100 core (1737–2000) and chlorophyll-a data (2002–
2012) does not overlap, and the overlap between MSA in the KC core and chlorophyll-a data is only five years, it was not
possible to perform correlations between chlorophyll-a and MSA in S100 or KC.

## 3 Results

### 3.1 MSA concentration and seasonal variability

The firn and ice core MSA records are shown in Figure 2. MSA concentrations are generally higher in the BI and KM cores
than in the KC and S100 cores. No significant trend is present in either of the cores for annual averaged MSA
concentrations. There is a significant autocorrelation for annual MSA anomalies in the KC core, but no further
autocorrelation was found in the BI, KM, or S100 cores.

Table 2 shows median and mean MSA concentration along with summer and winter mean MSA concentrations in the
four cores. Additionally, the results of the Wilcoxon rank-sum test for summer (ONDJFM) and winter (AMJJAS) mean
values are reported. Total median and mean MSA concentrations are approximately two times higher in the BI and KM cores
than in the KC and S100 cores. Summer mean MSA concentrations are also significantly higher in the BI and KM cores,
with 2- to 3-fold higher concentrations compared to the KC and S100 cores. There is a strong seasonal pattern evident in the





MSA record from the BI and KM cores, but the seasonality is not as clear in the KC or S100 core (Figure 4). The seasonal pattern in the BI and KM cores is supported by a statistically significant difference between summer and winter mean values

in the BI and KM cores, at a 95 % confidence level reported by the Wilcoxon rank-sum test. It is important to note that the accumulation and sampling resolution is higher in the BI and KM cores than in the KC and S100 cores, which could explain the lack of statistically significant difference between summer and winter MSA in the KC and S100 MSA records. The BI and KM MSA records display an MSA summer maximum (Figure 5). The winter to summer difference is largest in the BI core, which likely is an effect of higher accumulation as there is a southeast-northwest increase in snow accumulation at FIS

(Sinisalo et al., 2013). In the KC and S100 cores there are no statistically significant differences between summer and winter mean MSA values at a 95 % confidence level (Table 2).

**Table 2.** Median and mean total MSA concentration as well as seasonal mean MSA concentration in the BI, KC, KM, and S100 cores. Summer concentrations are based on ONDJFM and winter concentrations on AMJJAS. The results of the Wilcoxon rank-sum test are also reported.

| Core | Period (years) | Total median ($\mu$mol L$^{-1}$) | Total mean ($\mu$mol L$^{-1}$) | Summer mean ($\mu$mol L$^{-1}$) | Winter mean ($\mu$mol L$^{-1}$) | Sign. diff. winter and summer? | p-value |
|------|---------|---------|---------|---------|---------|---------|---------|
| **BI** | 1996–2012 | 0.36 | 0.47 | 0.71 | 0.24 | Yes | $1.21 \times 10^{-5}$ |
| **KC** | 1958–2007 | 0.16 | 0.21 | 0.19 | 0.23 | No | 0.09 |
| **KM** | 1995–2012 | 0.33 | 0.50 | 0.61 | 0.38 | Yes | 0.03 |
| **S100** | 1737–2000 | 0.12 | 0.17 | 0.17 | 0.17 | No | 0.52 |
| **S100** | 1989–2000 | 0.10 | 0.20 | 0.24 | 0.16 | No | 0.60 |

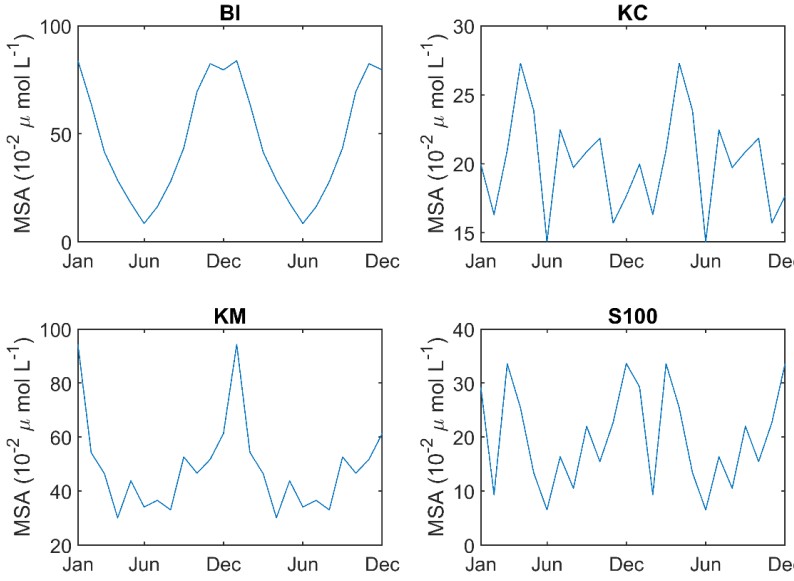


**Figure 4.** Seasonal variations in MSA concentration in the four cores depicted through monthly values averaged over the entire core length (BI, KC, and KM) or over the higher-resolution part of the core starting in 1989 (S100). The BI and KM cores show a pronounced





seasonality with high MSA values in the austral summer and low MSA in the austral winter, whereas at KC and S100 no significant seasonality is observed. The data are plotted over 24 months to display the seasonal pattern more clearly. The MSA data are monthly averages calculated off the raw MSA data.


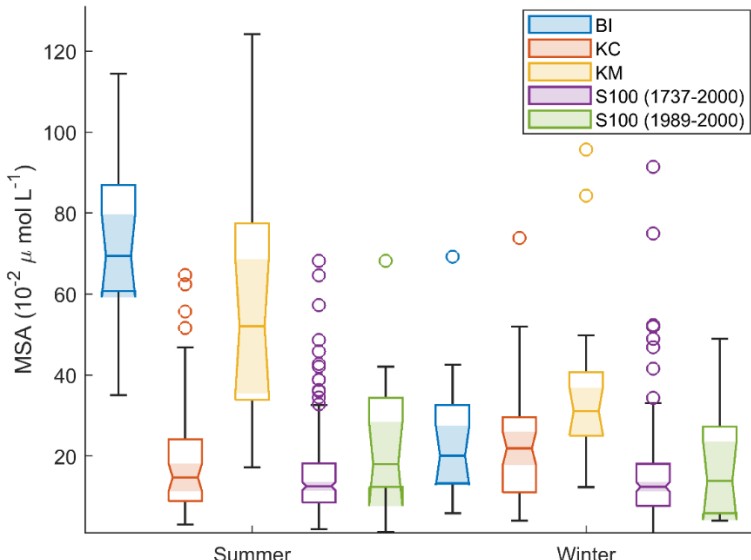

**Figure 5.** Seasonal MSA variability in the BI, KC, KM, and S100 cores. Summer concentrations were based on ONDJFM and winter concentrations on AMJJAS for the entire core (BI, KC, and KM) or for both the entire core and the period 1989–2000 (S100). The boxplots are based on the same populations as the Wilcoxon rank-sum test. The MSA data are seasonal averages calculated off the raw MSA data.


Normalised MSA in the form of calculated z-scores as well as the stacked MSA data are plotted in Figure 6 to Figure 8. The stack of seasonal MSA from the BI and KM cores show a general pattern of negative winter values and positive summer values but the winters of 2005 and 2006 stand out with positive values (Figure 6).





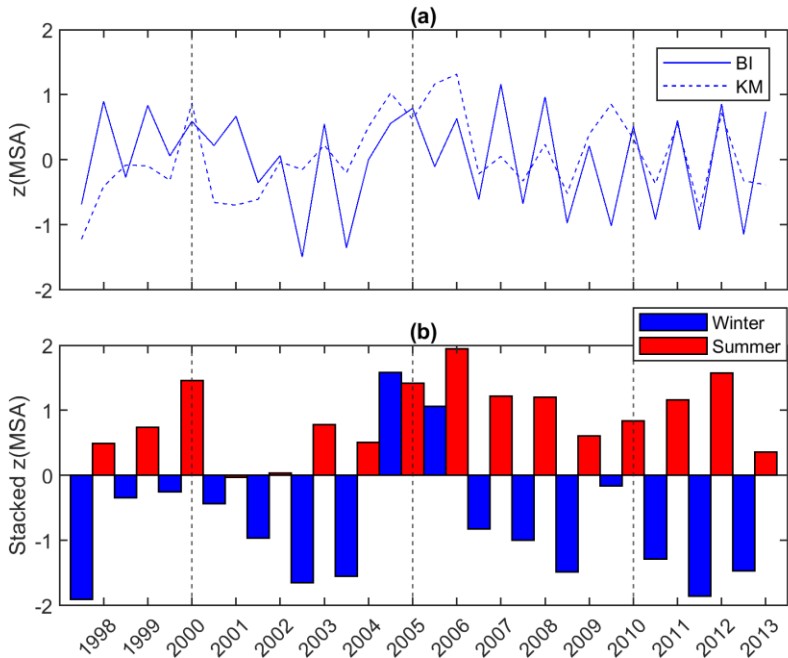


**Figure 6.** Summer and winter mean MSA normalised in the form of calculated z-scores. a) Individual BI and KM records and b) the stacked MSA data derived from the BI and KM MSA records.



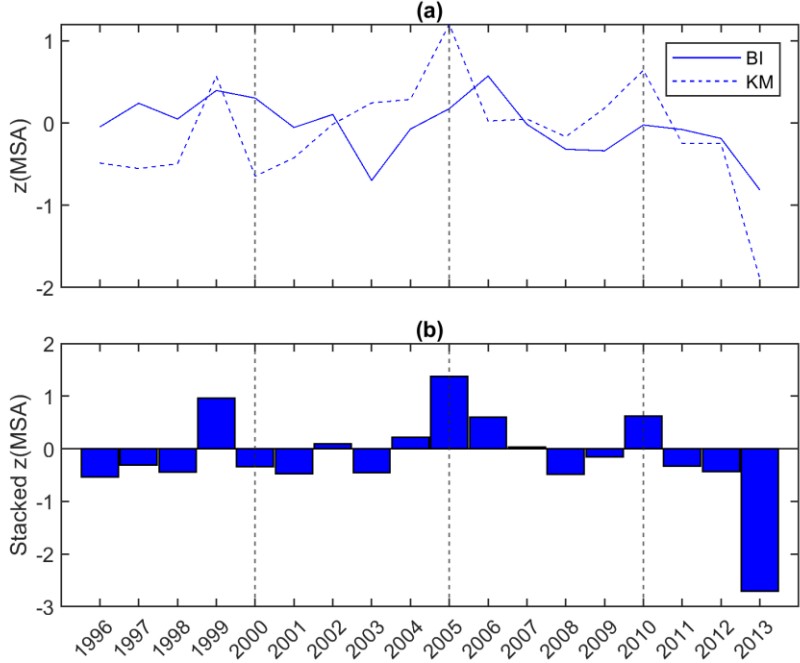

**Figure 7.** Annual mean MSA normalised in the form of calculated z-scores. a) Individual BI and KM records and b) the stacked MSA data derived from the BI and KM MSA records.



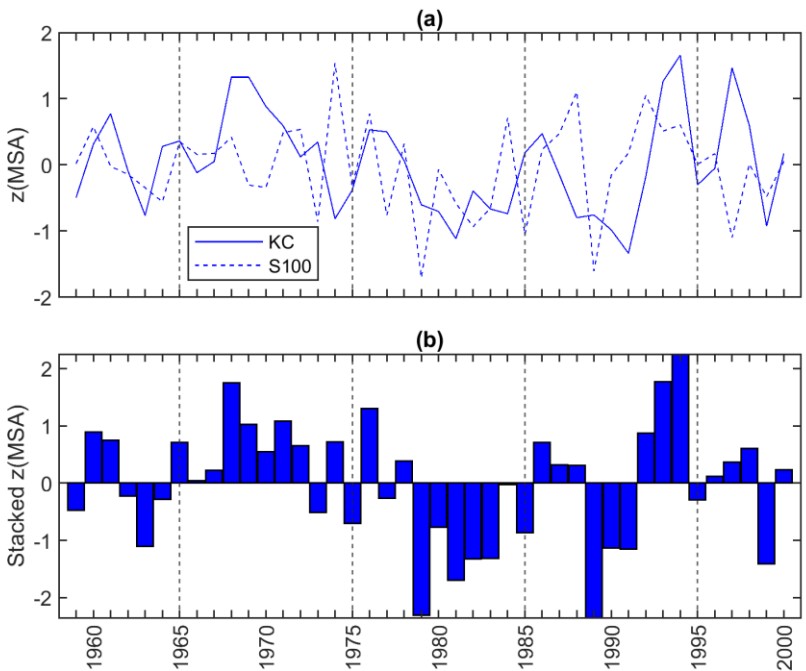

**Figure 8.** Annual mean MSA normalised in the form of calculated z-scores. a) Individual KC and S100 records and b) the stacked MSA data derived from the KC and S100 MSA records.

## 3.2 Trends in sea ice extent

Trends in winter averaged SIE for the five sea sectors and the Southern Ocean as a whole, for the years overlapping with the core coverages, are reported in Figure 9. Correlation coefficients and p-values for each fit show that there is a significant negative trend in the Bellingshausen-Amundsen Seas and significantly positive trends in the Indian Ocean, the Ross Sea, and the entire Southern Ocean, at the 95 % confidence level. However, there are no significant trends at the 95 % confidence level in the Weddell Sea Sector or the Western Pacific Ocean Sector.





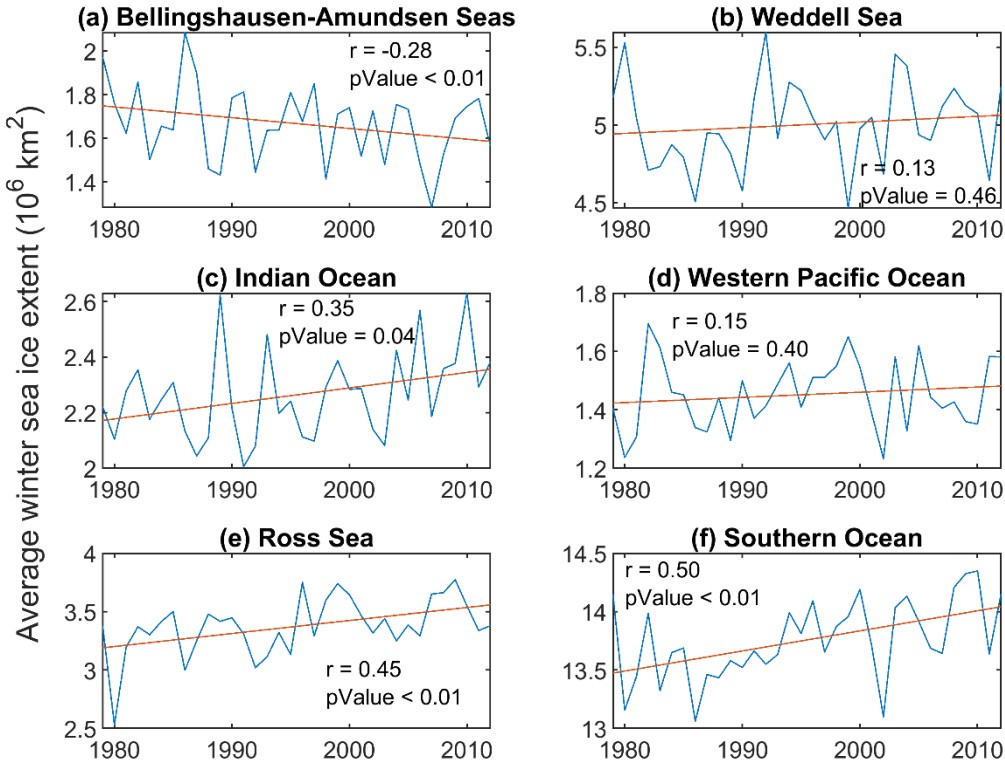

**Figure 9.** Winter averaged sea ice extents (SIE) for January 1979 – December 2012 with corresponding trend lines, correlation coefficients, and p-values in a. Bellingshausen-Amundsen Seas, b. Weddell Sea, c. Indian Ocean, d. Western Pacific Ocean, e. Ross Sea, and f. the entire Southern Ocean.

**3.3 Correlation analysis**

**3.3.1 MSA – SIE**

Table 3 presents the results of the MSA – SIE correlation analysis for correlation coefficients significant at a 95 % confidence level. They show a positive significant correlation for the Weddell Sea Sector winter SIE with KM summer MSA. A negative correlation of comparable strength is present for the Western Pacific Sea Sector with summer MSA in the

KM core. There is also a significant positive correlation between Weddell Sea Sector winter SIE and stacked BI and KM summer MSA. Furthermore, annual MSA stacked from the KC and S100 cores shows a statistically significant negative correlation to winter SIE in both the Indian Ocean Sector and the Ross Sea Sector. Results for all sea sector and core combinations are reported in the appendix (Table A5 – Table A7).






**Table 3.** Coefficients of determination between MSA and SIE are reported together with their respective p-values for the sea sector – core combinations that are significant at a 95 % confidence level.

| Winter SIE | KM seasonal | | BI + KM seasonal stack | | KC + S100 annual stack | |
|---|---|---|---|---|---|---|
| | $R^2$ | p-value | $R^2$ | p-value | $R^2$ | p-value |
| **Weddell Sea Sector** | 0.32 (positive) | 0.01 | 0.38 (positive) | < 0.01 | - | - |
| **Western Pacific Ocean Sector** | 0.28 (negative) | 0.02 | - | - | - | - |
| **Indian Ocean Sector** | - | - | - | - | 0.27 (negative) | 0.01 |
| **Ross Sea Sector** | - | - | - | - | 0.19 (negative) | 0.03 |

### 3.3.2 MSA – chlorophyll-a

The summer MSA record in the BI core shows a significant positive correlation to chlorophyll-a concentration in the Weddell Sea Sector under both short-range transport (25 km buffer zone) and long-range transport (100 km buffer zone) of MSA at a 95 % confidence level (Table 4). Additionally, the annual MSA record in the BI core is significantly positively correlated to chlorophyll-a concentrations in the Ross Sea Sector under both short- and long-range transport of MSA. Finally, the stacked BI and KM summer MSA is significantly positively correlated to chlorophyll-a concentration in the Ross

Sea Sector under short-range and long-range transport of MSA. For the remaining cores and sea sectors combinations, no significant correlations at a 95 % confidence level are reported (Table A8 – Table A10).

**Table 4.** Coefficients of determination between MSA and chlorophyll-a are reported together with their respective p-values for the sea sector – core combinations that are significant at a 95 % confidence level.

| Summer chlorophyll-a | | BI seasonal | | BI annual | | BI + KM seasonal stack | |
|---|---|---|---|---|---|---|---|
| | | $R^2$ | p-value | $R^2$ | p-value | $R^2$ | p-value |
| **Weddell Sea Sector** | **25 km buffer zone** | 0.39 (positive) | 0.03 | - | - | - | - |
| | **100 km buffer zone** | 0.39 (positive) | 0.03 | - | - | - | - |
| **Ross Sea Sector** | **25 km buffer zone** | - | - | 0.48 (positive) | 0.02 | 0.35 (positive) | 0.04 |





| | | | 0.44 (positive) | 0.02 | 0.32 (positive) | 0.05 |
|---|---|---|---|---|---|---|
| **100 km buffer zone** | - | - | 0.44 (positive) | 0.02 | 0.32 (positive) | 0.05 |

## 4 Discussion

### 4.1 Post-depositional processes

Post-depositional processes such as diffusion and percolation of MSA may disturb the initially deposited MSA profile with implications for conclusions one can draw from the analysis of the data (Mulvaney et al., 1992; Curran et al., 2002; Fundel et al., 2006). To investigate potential post-depositional MSA relocation at the study sites, the seasonality of major ions in the cores have been studied in previous work by Vega et al. (2016). All four cores used in this study contain ice layers, which indicate episodes of melt. In the KM and BI cores, the $\delta^{18}O$ and major ion records are well preserved with a pronounced seasonality indicating that events of melt and percolation were not intensive enough to disturb the MSA signal (Vega et al. 2016). The stratigraphic record in the KC core does not exhibit as clear seasonal cycles as in the KM and BI cores, but no relationship was found when comparing the ice layers in the core with MSA anomalies (Vega et al. 2016). Thus, it is not likely that any significant mass transport and redistribution of ions via percolation has occurred at the KC site either. The absence of seasonal cycles is therefore more likely a result of low accumulation and post-depositional processes such as wind scouring (Vega et al. 2016), which does not prevent the MSA record from being reliably compared to climate parameters such as sea ice on an interannual scale. In the S100 core, ice layers are thicker than in the ice rises cores (Kaczmarska et al. 2006) which, given the lower accumulation in S100, indicates a larger melt influence at this low elevation site. It can thus not be ruled out that melt and percolation could have affected the MSA signal at this site.

### 4.2 Resolution of the MSA records

Due to lack of significant differences between summer and winter mean MSA concentration in the KC and S100 cores (Table 2), and the lack of summer maxima in the KC MSA record (Figure 5), which is expected because of the relatively higher DMS production in the SIZ during spring and summer (Kettle et al., 1999; Curran and Jones, 2000; Hezel et al., 2011), we do not perform correlation analysis on the seasonal MSA in the KC and S100 cores. Thus, only annual MSA averages of the KC and S100 cores were used in the correlation analysis.

### 4.3 Relationship between MSA and sea ice extent

The KM summer MSA record is significantly positively correlated to SIE in the Weddell Sea Sector. There is also a significant positive correlation between stacked BI – KM summer MSA and SIE in the Weddell Sea Sector. In contrast, the BI core MSA record is not significantly correlated to SIE in any sea sector. Vega et al. (2018) found that the fluxes of sea salt related ions (Cl⁻, Na⁺, Mg²⁺, and Ca²⁺) to the BI, KM, KC, and S100 core sites differ. KM generally receives higher





concentrations of all sea salt species, both in summer and winter, than BI and KC. Furthermore, summer sea salt concentrations in BI are lower than winter concentrations for at least three of the sea salt species. These differences in sea salt load were explained by wind and precipitation patterns on FIS. The main source of precipitation is onshore flow of

easterly and north-easterly winds, caused by eastward cyclone movement north of the coast (Schlosser et al., 2008). Due to this wind regime, the KM ice rise lies closest to the marine source of major ions, whereas efficient transport of sea salt ions to the BI site is hampered by about 150 km transport pathway over land despite being situated only 10 km from the coast (Figure 1). The KC core is also located farther away from the source of major sea salt ions at 42 km distance from the coast. When travelling over land, exchange processes between the air masses and snow, such as deposition of the ion load, affects

the sea salt signal at the core location (Vega et al., 2018). It could also be a case of fog formation altering the signal at the inland sites, which is likely not an equally important problem at the KM site because of stronger winds. These processes might also affect the delivery of MSA to the BI ice rise, causing the signal to be different from that of KM. Even though the total median and mean MSA concentrations in the BI and KM cores are similar, the delivery of MSA to the core sites on a temporal basis might be affected by these processes, resulting in the different correlation patterns observed in the two cores.

Both field- and model-based studies show that MSA deposition decreases with increasing distance inland (Becagli et al., 2005; Hezel et al., 2011). However, to establish the exact processes occurring during the travel from the coastal KM to the BI site, or from the marine source to the more inland KC site, data from meteorological stations at FIS would be needed.

In the KC and S100 stacked annual MSA data we find a negative correlation to winter SIE in both the Indian Ocean Sector and the Ross Sea Sector. This is unexpected as we have no evidence for significant delivery of MSA produced in the

Indian Ocean or Ross Sea Sector to our core sites. The overall difference in correlation patterns between the MSA records from our four core sites could be explained by the fact that, despite the relative proximity of the core sites, different meteorological conditions are prevalent at each site. As seen in other isotope data from the cores, there is a difference in delivery to the core sites, particularly if comparing the dome-shaped ice rises BI and KM to the more elongated ice rise KC or the level S100 site (Vega et al. 2016, 2018).

### 4.3.1 The Western Pacific Ocean Sector

Since the Weddell Sea Sector is the general source region for moisture to DML (Noone et al. 1999; Reijmer and Broeke 2001; Schlosser et al. 2008), a significant correlation between MSA and SIE in the Weddell Sea Sector was expected. Furthermore, the correlation was expected to be positive, assuming that a larger SIE leads to more marine primary productivity, which in turn increases the production of MSA. In the Western Pacific Ocean Sector, a negative correlation

was found for winter SIE with summer MSA averages in the KM core. It is unlikely that a significant amount of MSA produced in the Western Pacific Ocean Sector is transported to the core sites around Antarctica. The negative correlation found to the MSA record in the KM core might instead reflect an intermittent teleconnection between the sea ice conditions in the Weddell Sea and the Western Pacific Ocean, where years with larger SIE in the Weddell region are accompanied by a smaller SIE in the Western Pacific. Spearman's rank correlation was applied to the Weddell and Western Pacific SIE for the



period overlapping the KM core (1995–2012) to investigate this further. A negative correlation between the sea sectors was indeed found. Although not significant at the 95 % confidence level, the fact that the same negative correlation was not visible when comparing the entire record of SIE (1979–2021) in the Weddell and Western Pacific point to a possible connection between sea ice conditions in these sea sectors during the years corresponding to the coverage of the KM core.

### 4.4 Relationship between MSA and chlorophyll-a

A significant positive correlation between MSA and chlorophyll-a in the Weddell Sea Sector, similar to the one we find for the BI summer MSA record under both short-range and long-range MSA transport, could point towards the Weddell Sea region being the main source of MSA to FIS. However, we see no significant correlation between chlorophyll-a in the Weddell Sea Sector and MSA in the KM core, where significant MSA – SIE correlations were found. Thus, the support for that hypothesis weakens. Furthermore, we also find significant correlations between chlorophyll-a concentrations in the Ross

Sea Sector and MSA in the annual BI MSA record as well as in the stacked BI – KM MSA record. Since we do not expect significant transport of MSA produced in the Ross Sea Sector to the core sites at FIS we can conclude that there is no evidence to connect the chlorophyll-a concentration in the Southern Ocean to MSA in FIS firn and ice cores.

   Generally, the results of the MSA – chlorophyll-a correlation analysis should be interpreted with caution because of the short overlapping period between the MSA and chlorophyll-a data. This could explain the lack of correlation for the KM

MSA record to chlorophyll-a when significant correlations were found between MSA in the BI core and SIE.

   Another issue is the approach taken when limiting the spatial extent of the chlorophyll-a data. When doing this, the September sea ice edge for 2002–2012 was used, which means that all months of chlorophyll-a data were limited using the position of the September sea ice edge. The resulting chlorophyll-a mean concentration for e.g. January will then include areas of the ocean that are located far north of the influence of the actual sea ice edge in January, which could alter the

signal.

### 4.5 Suitability of the MSA records for sea ice reconstruction

   The correlations identified between FIS MSA firn and ice core records and SIE and chlorophyll-a are relatively weak. In addition, the two cores with better MSA preservation, BI and KM, do not display the same correlation patterns between MSA and SIE or MSA and chlorophyll-a. These findings do not favour using the FIS MSA records to reconstruct sea ice.

Core stacking can be helpful to enhance the MSA signal and reduce noise arising from, for example, varying snow fall at separate sites (Abram et al., 2007). The firn and ice cores used in this study are influenced by different meteorological patterns affecting snow deposition (Lenaerts et al. 2014; Vega et al. 2016), so the use of stacked cores is warranted. Stacking the MSA record from multiple cores together before correlation analysis did reveal significant correlations to the stacked MSA record from the KC – S100 cores, where without the stacking we did not identify any significant correlations to neither

the KC nor the S100 MSA records. However, no clear pattern between the MSA – SIE correlations and the MSA –





chlorophyll-a correlations arose after core stacking, further showing the unsuitability of the different FIS MSA records to consistently reconstruct sea ice.

In this study, the easterly to north-easterly winds transporting MSA onshore FIS have a large influence on the transport pathway of MSA, in turn influencing the MSA signal in the cores. A detailed modelling study of atmospheric transport and
dispersion of MSA at FIS, which is beyond the scope of this paper, might help to improve the understanding of the physical reasons for the correlations or lack thereof between MSA in the cores and SIE /chlorophyll-a presented in our study.

## 5 Conclusions

Here we investigate the MSA records from three firn cores (BI, KM, KC) and one ice core (S100) drilled in Fimbul Ice Shelf in Dronning Maud Land, East Antarctica as potential proxy of sea ice extent (SIE) in the Southern Ocean. For this, MSA
records were correlated with satellite-derived SIE and chlorophyll-a concentrations as a measure of phytoplankton biomass. We find no clear correlation pattern between the MSA records across the four investigated cores and SIE or chlorophyll-a in the Southern Ocean. The significant positive correlation for summer MSA in the KM core and summer MSA in the BI – KM stack to winter SIE in the Weddell Sea Sector, as well as the significant positive correlation between summer MSA in the BI core and summer chlorophyll-a concentrations in the Weddell Sea Sector could point to a reflection of the main source
region of moisture and therefore also MSA to Dronning Maud Land being the Weddell Sea Sector. However, we find no significant correlation between MSA in the BI core to SIE in the Weddell Sea Sector and no significant correlation between MSA in the KM core to chlorophyll-a in the Weddell Sea Sector to corroborate it. The use of stacked MSA records in the correlation calculations introduced significant correlations to SIE and chlorophyll-a in the Indian Ocean and Ross Sea Sectors, as well as showed an additional positive correlation to winter SIE in the Weddell Sea Sector. The lack of coherent
correlations of MSA to SIE or chlorophyll-a, even with the stacked MSA records, shows the unsuitability of the FIS MSA records to reconstruct past sea ice conditions in the Southern Ocean.

## 6 Data availability

The MSA records of KM, BI, KC, and S100 cores are available for download from PANGEA at
https://doi.org/10.1594/PANGAEA.889018 (Vega et al., 2022). Sea ice extent data can be accessed via the National Snow
and Ice Data Center (https://doi.org/10.7265/N5K072F8, Fetterer et al., 2022). Chlorophyll-a concentration data set is available at doi:10.5067/AQUA/MODIS/L3M/CHL/2018 (NASA Goddard Space Flight Center, 2022). MODIS Mosaic of Antarctica 2013-2014 image can be found at https://doi.org/10.5067/RNF17BP824UM (Haran et al., 2022) and the data for imaging the Antarctic coastline can be accessed via https://doi.org/10.5285/ed0a7b70-5adc-4c1e-8d8a-0bb5ee659d18 (Gerrish et al., 2022).



## Author contribution

The paper is a rewrite of the MSc thesis of first author EN (Nilsson, 2022). EI has been the PI for the field activities, coordinated the analysis and interpretations. RM, AE, and MS provided the ion analysis. TM analysed the water isotopes. CPV designed the original study behind this paper. DD guided the quantitative analysis. EN prepared the manuscript with contributions from all co-authors.

## Competing interests

Some authors are members of the editorial board of Ice core science at the three poles (CP/TC inter-journal SI).

## Acknowledgements

We are grateful to those who helped to collect, transport, sample, and analyse the firn cores from FIS. Financial support for these projects came from the Norwegian Research Council and the Norwegian Polar Institute.

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





**Appendix**

**Table A5.** Results for correlation calculations between summer MSA and winter SIE. Coefficient of determination (adjusted) is reported along with the p-value for each sea sector and core combination. Correlations are based on summer as ONDJFM and winter as AMJJAS. Cells highlighted in green represent significant correlations at a 95 % significance level.

| | BI seasonal | | KM seasonal | |
|---|---|---|---|---|
| | $R^2$ | p-value | $R^2$ | p-value |
| Weddell Sea | -0.05 | 0.91 | 0.32 positive | 0.01 |
| Western Pacific Ocean | -0.01 | 0.41 | 0.28 negative | 0.02 |
| Bellingshausen-Amundsen Seas | -0.01 | 0.67 | 0.04 | 0.23 |
| Indian Ocean | -0.03 | 0.47 | -0.02 | 0.44 |
| Ross Sea | 0.09 | 0.16 | -0.02 | 0.42 |
| Circum-Antarctica | -0.06 | 0.78 | -0.02 | 0.42 |

**Table A6.** Results for correlation calculations between annual MSA and winter SIE. Coefficient of determination (adjusted) is reported
along with the p-value for each sea sector and core combination. Correlations are based on winter as AMJJAS. Annual anomalies are based on July to June values. Cells highlighted in green represent significant correlations at a 95 % significance level.

| | BI annual | | KM annual | | KC annual | | S100 annual | |
|---|---|---|---|---|---|---|---|---|
| | $R^2$ | p- value | $R^2$ | p- value | $R^2$ | p- value | $R^2$ | p- value |
| Weddell Sea | -0.04 | 0.62 | 0.05 | 0.30 | -0.01 | 0.42 | -0.03 | 0.50 |
| Western Pacific Ocean | 0.03 | 0.29 | 0.10 | 0.11 | -0.03 | 0.74 | -0.05 | 0.80 |
| Bellingshausen-Amundsen Seas | -0.05 | 0.78 | 0.13 | 0.09 | -0.01 | 0.42 | 0.04 | 0.20 |
| Indian Ocean | -0.04 | 0.75 | -0.05 | 0.81 | -0.02 | 0.26 | -0.02 | 0.45 |
| Ross Sea | -0.01 | 0.46 | -0.04 | 0.64 | 0.06 | 0.11 | 0.17 | 0.04 |
| Circum-Antarctica | -0.04 | 0.59 | -0.06 | 0.90 | 0.10 | 0.06 | -0.02 | 0.26 |





**Table A7.** Results for correlation calculations between 1. stacked summer MSA in the BI and KM cores and winter SIE and 2. Stacked annual MSA in the BI + KM cores and KC + S100 cores and winter SIE. Coefficient of determination (adjusted) is reported along with the p-value for each sea sector and core combination. Correlations are based on summer as ONDJFM and winter as AMJJAS. Cells highlighted in green represent significant correlations at a 95 % significance level.

| | BI + KM seasonal | | BI + KM annual | | KC + S100 annual | |
|---|---|---|---|---|---|---|
| | $R^2$ | p- value | $R^2$ | p- value | $R^2$ | p- value |
| Weddell Sea | 0.38 positive | < 0.01 | 0.02 | 0.29 | -0.06 | 0.97 |
| Western Pacific Ocean | 0.15 | 0.08 | -0.04 | 0.95 | -0.05 | 0.77 |
| Bellingshausen-Amundsen Seas | -0.05 | 0.65 | 0.00 | 0.45 | 0.01 | 0.30 |
| Indian Ocean | 0.10 | 0.13 | 0.02 | 0.36 | 0.27 negative | 0.01 |
| Ross Sea | 0.09 | 0.14 | -0.04 | 0.97 | 0.19 negative | 0.03 |
| Circum-Antarctica | 0.00 | 0.35 | 0.01 | 0.41 | -0.05 | 0.71 |

**Table A8.** Results for correlation calculations between summer MSA and chlorophyll-a for both a 25 km buffer zone and a 100 km buffer zone. Coefficient of determination (adjusted) is reported along with the p-value for each sea sector and core combination. Correlations are based on summer MSA as ONDJFM, and summer chlorophyll-a as ONDJFM. Cells highlighted in green represent significant correlations at a 95 % significance level.

| | | BI seasonal | | KM seasonal | |
|---|---|---|---|---|---|
| | | $R^2$ | p-value | $R^2$ | p-value |
| Weddell Sea | 25 km buffer zone | 0.39 positive | 0.03 | -0.01 | 0.41 |
| | 100 km buffer zone | 0.39 positive | 0.03 | -0.12 | 1.00 |
| Western Pacific Ocean | 25 km buffer zone | 0.07 | 0.23 | -0.11 | 0.81 |
| | 100 km buffer zone | 0.07 | 0.23 | -0.11 | 0.78 |
| Bellingshausen-Amundsen Seas | 25 km buffer zone | -0.06 | 0.49 | -0.12 | 0.88 |
| | 100 km buffer zone | -0.06 | 0.50 | -0.12 | 0.87 |
| Indian Ocean | 25 km buffer zone | -0.12 | 0.85 | 0.17 | 0.15 |
| | 100 km buffer zone | -0.12 | 0.86 | 0.17 | 0.14 |
| Ross Sea | 25 km buffer zone | 0.23 | 0.09 | -0.08 | 0.59 |
| | 100 km buffer zone | 0.22 | 0.10 | -0.08 | 0.56 |





| | | | | | |
|---|---|---|---|---|---|
| Circum-Antarctica | 25 km buffer zone | 0.14 | 0.15 | -0.09 | 0.61 |
| | 100 km buffer zone | 0.14 | 0.16 | -0.08 | 0.60 |

**Table A9.** Results for linear correlation calculations between annual MSA and chlorophyll-a for both a 25 km buffer zone and a 100 km buffer zone. Coefficient of determination (adjusted) is reported along with the p-value for each sea sector and core combination. Correlations are based on annual MSA as July to June, and summer chlorophyll-a as ONDJFM. Cells highlighted in green represent significant correlations at a 95 % significance level.

| | | BI annual | | KM annual | |
|---|---|---|---|---|---|
| | | $R^2$ | p-value | $R^2$ | p-value |
| Weddell Sea | 25 km buffer zone | 0.23 | 0.09 | -0.09 | 0.70 |
| | 100 km buffer zone | 0.19 | 0.12 | -0.10 | 0.73 |
| Western Pacific Ocean | 25 km buffer zone | -0.04 | 0.55 | -0.10 | 0.67 |
| | 100 km buffer zone | -0.04 | 0.53 | -0.10 | 0.70 |
| Bellingshausen-Amundsen Seas | 25 km buffer zone | -0.12 | 0.84 | -0.10 | 0.96 |
| | 100 km buffer zone | -0.11 | 0.83 | -0.10 | 0.73 |
| Indian Ocean | 25 km buffer zone | -0.12 | 0.86 | -0.12 | 0.90 |
| | 100 km buffer zone | -0.12 | 0.84 | -0.11 | 0.90 |
| Ross Sea | 25 km buffer zone | 0.48 Positive | 0.02 | -0.06 | 0.51 |
| | 100 km buffer zone | 0.44 positive | 0.02 | -0.05 | 0.49 |
| Circum-Antarctica | 25 km buffer zone | 0.10 | 0.20 | 0.11 | 0.22 |
| | 100 km buffer zone | 0.09 | 0.21 | 0.12 | 0.21 |

**Table A10.** Results for linear correlation calculations between seasonal and annual MSA stacked from the individual records in the BI and KM cores and chlorophyll-a for both a 25 km buffer zone and a 100 km buffer zone. Coefficient of determination (adjusted) is reported along with the p-value for each sea sector and core combination. Correlations are based on annual MSA as July to June, and summer chlorophyll-a as ONDJFM. Cells highlighted in green represent significant correlations at a 95 % significance level.

| | | BI + KM seasonal stack | | BI + KM annual stack | |
|---|---|---|---|---|---|
| | | $R^2$ | p-value | $R^2$ | p-value |
| Weddell Sea | 25 km buffer zone | 0.04 | 0.27 | -0.05 | 0.86 |
| | 100 km buffer zone | 0.05 | 0.26 | -0.05 | 0.82 |



| | | | | | |
|---|---|---|---|---|---|
| Western Pacific Ocean | 25 km buffer zone | 0.16 | 0.15 | 0.14 | 0.27 |
| | 100 km buffer zone | 0.16 | 0.15 | 0.13 | 0.27 |
| Bellingshausen-Amundsen Seas | 25 km buffer zone | -0.09 | 0.63 | 0.27 | 0.13 |
| | 100 km buffer zone | -0.09 | 0.64 | 0.28 | 0.13 |
| Indian Ocean | 25 km buffer zone | -0.09 | 0.62 | 0.02 | 0.54 |
| | 100 km buffer zone | -0.08 | 0.60 | 0.01 | 0.55 |
| Ross Sea | 25 km buffer zone | 0.35 positive | 0.04 | 0.05 | 0.56 |
| | 100 km buffer zone | 0.32 positive | 0.05 | 0.06 | 0.54 |
| Circum-Antarctica | 25 km buffer zone | 0.26 | 0.08 | < 0.01 | 0.63 |
| | 100 km buffer zone | 0.26 | 0.08 | 0.02 | 0.59 |