# Peer review of "Methane Sulphonic Acid in East Antarctic Coastal Firn and Ice Cores and Its Relationship with Chlorophyll-a and Sea Ice Extent"

_EGUsphere, 2023_

## Referee Comment (RC1)

**Review for:**

**Methane Sulphonic acid in East Antarctic Coastal Firn and Ice Cores and Its Relationship with Chlorophyll-a and Sea Ice Extent**

Nilsson et al., 2024

**Summary**

The article interprets MSA records from three firn cores and one ice core from Fimbul ice shelf in the East Antarctic core. MSA (methanesulphonic acid) concentration in the atmosphere is linked to oceanic primary productivity for this reason it has been used in several studies to reconstruct past Antarctic sea ice variability. These studies however show contrasting results and at present the abundance of this compound in Antarctic snow can't be used confidently for past sea ice reconstructions.

The aim of this manuscript is to determine whether MSA concentrations in firn and ice cores from the Fimbul ice shelf are linked to sea ice variability and chlorophyll. The study finds some positive (although weak) correlation with both sea ice and chlorophyll in the Weddell sea sector, however, the lack of coherency in the correlations across all cores evidences the unsuitability of the site for past sea ice reconstructions using MSA.

**General comments**

I think the article is well written and with good level of English. I appreciated the discussion on the differences among the sites resulting in big differences in MSA deposition even though the sites are quite close to each other. Maybe this is something that could be highlighted better in the manuscript.

However, I think there are several major weaknesses in this article that should be addressed before publication. First and most important, I struggle to see the novelty in this study. There have been several publications of MSA in Antarctica showing contrasting results as the authors also state in the Introduction but I feel the study lacks an in-depth discussion on why the site is unsuitable for sea ice reconstructions using MSA (atmospheric patterns? Orography? Ocean circulation?). I would also apreciate if the authors could add a discussion on how this finding relates to these previous studies. This discussion would help defining a logic for determining whether a site is suitable or not for sea ice reconstructions using MSA.

Second, the manuscript is not clear whether the aim of the study is to investigate the MSA-sea ice relationship across the entire Southern ocean or in the source area of impurities uplifted, transported and deposited in the Fimbul ice shelf. Studies using sea ice proxies (e.g. Thomas et al., 2019) define the source area of impurities transported the site, so in this context showing correlations with all sectors of the Southern ocean doesn't make too much sense. I think the authors should at least state which of the sectors are the ones that should correlate the most given the proximity and the air mass transport. Defining the source area could be done through backtrajectory analysis or by considering main wind patterns and the lifetime of MSA in the atmosphere.

Third, the manuscript shows quite weak correlations that are computed on fairly large sectors. How can it be argued that the identified correlations are not merely spurious? Here the authors are comparing multiple records against multiple sectors of the Southern Ocean, so there is a fair amount of chance that some of the correlations will be positive, even if no relation exists. Can the authors exclude that this is the case?

**Specific comments**

Line 26-28: In this sentence the authors first refer the sea ice in the Southern ocean and then to MSA being a proxy of regional sea ice. Please modify the sentence to be consistent to either hemispheric or regional sea ice.

Line 31-32: please explain how sea ice is a facilitator of DMS production.

Line 38: MSA is not typically used for reconstructing sea ice in Arctic ice cores. Please see Osman et al., (2019). In their study they use MSA records from the Greenland ice sheet to reconstruct subarctic productivity that changed in relation to oceanic circulation, rather than Arctic sea ice decline in the industrial era.

Line 59: "There are now longer records": with respect to which ones specifically?

Line 65: "and chlorophyll-a concentrations IN the Southern Ocean"

Line 104: can you describe the thickness of the layers?

Line 137: I cannot find this citation in the bibliography

Figure 5: please make clearer that the first 5 boxplots are for summer and the other 5 are for winter. My suggestion is two curly brackets which contain summer and winter boxplots.

Line 247-248: please move up "from the BI and KM cores"

Line 260: this section diverts from the scope of the study I suggest to remove it

Lines 331-332: Please state why it is still consistent to stack them together

Lines 397-400: can you split up this sentence? It is very long to read

Lines 402-404: I don't think that positive correlations in the Ross sea sector is of any significance as it's outside the source area of moisture transported to the Fimbul ice shelf. I also find hardly significant the correlation found in the Indian ocean given that westerly winds transport air masses eastward (see for instance Fig 2 of Clem et al., 2020). Please acknowledge this when stating that MSA records correlated with sea ice and chlorophyll in this sectors.

**Bibliography**

Clem, K. R., Fogt, R. L., Turner, J., Lintner, B. R., Marshall, G. J., Miller, J. R., & Renwick, J. A. (2020). Record warming at the South Pole during the past three decades. *Nature Climate Change*, *10*(8), 762–770. https://doi.org/10.1038/s41558-020-0815-z

Osman, M. B., Das, S. B., Trusel, L. D., Evans, M. J., Fischer, H., Grieman, M. M., Kipfstuhl, S., McConnell, J. R., & Saltzman, E. S. (2019). Industrial-era decline in subarctic Atlantic productivity. *Nature*, *569*(7757), 551–555. https://doi.org/10.1038/s41586-019-1181-8

Thomas, E. R., Allen, C. S., Etourneau, J., King, A. C. F., Severi, M., Winton, V. H. L., Mueller, J., Crosta, X., & Peck, V. L. (2019). Antarctic sea ice proxies from marine and ice core archives suitable for reconstructing sea ice over the past 2000 years. *Geosciences (Switzerland)*, *9*(12). https://doi.org/10.3390/geosciences9120506